# Fertility Preservation and Long-Term Monitoring of Gonadotoxicity in Girls, Adolescents and Young Adults Undergoing Cancer Treatment

**DOI:** 10.3390/cancers13020202

**Published:** 2021-01-08

**Authors:** Kaja Michalczyk, Aneta Cymbaluk-Płoska

**Affiliations:** Department of Gynecological Surgery and Oncology of Adults and Adolescents, Pomeranian Medical University, al. Powstańców Wielkopolskich 72, 70-111 Szczecin, Poland; anetac@data.pl

**Keywords:** oncology, fertility, chemotherapy, radiotherapy, gynecology, cryopreservation, survivorship, AYA

## Abstract

**Simple Summary:**

As cancer treatments become more effective and many patients have long term survival, concerns related to patient’s quality of life and reproductive health become relevant. It is especially important for girls and young females facing cancer therapy who have not yet started family planning. This review provides insight into current fertility preservation methods for pre- and post-pubertal girls and young adults undergoing cancer treatment. It contributes to this research area by providing evidence-based information on currently used methods, follow-up and survivorship care. Ethical considerations related to oncofertility in pediatric and adolescent patients were raised. Psychological aspects and possible issues that may occur at different timing (diagnosis, fertility preservation and follow-up) were evaluated.

**Abstract:**

Chemo- and radio-therapy can often affect reproductive organs impairing hormonal regulation, fertility, and sexual function. As cancer treatments become more effective and many patients have long term survival, concerns related to patient’s quality of life and reproductive health become relevant. It is especially important for girls and young females facing cancer therapy who have not yet started family planning. Chemotherapy protocols using alkylating agents and abdominal radiotherapy, which are frequently used in the treatment of childhood and adolescent cancer, can cause gonadal injury. The most common clinical manifests are ovarian hormone insufficiency, premature ovarian insufficiency, early menopause and infertility. In this review we assess current literature and summarize current recommendations on the reproductive function of girls and young females undergoing cancer treatment and their follow-up. Fertility preservation methods are discussed, including psychological and ethical considerations and barriers. Improvement of reproductive health and quality of life of adolescents and young adults (AYA) undergoing cancer treatment is an important issue. Further research should be continued to develop efficient and accessible methods for fertility preservation in young patients. An expert panel including oncologists, radiation oncologists, endocrinologists and gynecologists should always consider fertility preservation in pediatric, adolescent and AYA cancer patients, minding patients’ medical condition, cancer staging and potential risk of treatment-related gonadotoxicity.

## 1. Introduction

Childhood cancer survival has improved greatly over the last decades with pediatric cancer survival rates exceeding 80% [1]. Therapies commonly used in childhood and adolescent cancer treatment including chemotherapy agents and irradiation of CNS (central nervous system) and/or pelvis can affect pubertal development, hormonal regulation and alter fertility reducing the quality of later life. Adolescents and young adults (AYA) undergoing cancer treatment can expect good prognosis and long-term survival, often similar to non-cancer populations, as approximately 80% of AYAs will achieve long-term cure. However, there are multiple issues that need to be considered in order to improve the quality of life of such patients. Patients should be monitored and provided with proper survivorship care upon the end of cancer treatment, in order to allow them good psychosocial functioning.

There are some special issues that should be considered before the beginning of cancer treatment of adolescent and AYA patients, one of them being the long-term and late effects of the treatment, that many forget about once fearing for their life. Gonadal dysfunction and infertility are major concerns of patients and their families, causing additional fear and anxiety related to cancer treatment [2,3,4]. With improved childhood and adolescent cancer survival rates comes a growing population of survivors who are or will be interested in having their own children. Upon having an option to undergo fertility preservation treatment, many young patients are willing to go forward with the procedures.

It should be emphasized that most of the early complications such as gastric disorders, alopecia and weakness are transient. Despite causing a bigger problem, late complications—including infertility and secondary cancers—are often forgotten as they are delayed in time and may appear after many years. The long-term effects of therapy and patient’s quality of life should be considered both before and after treatment. Upon considerations of treatment-related risk factors, patients counseling and referral to a specialist can help facilitate further patient management to preserve fertility in high-risk patients [5]. Options for fertility preservation in prepubertal patients are few and still considered experimental.

## 2. Agents Affecting Fertility

The risk of reproductive complications is affected by the type of treatment, its intensity and modality, age at diagnosis as well as a primary diagnosis and the site of disease [6,7]. Regimens consisting of chemotherapy and pelvic radiotherapy can cause adverse effects on gonadal function leading to a dose-related decline of ovarian reserve [8]. As reported by long term follow-up studies of childhood cancer survivors, about 8% develop premature ovarian failure (POI). If chemotherapy is combined with radiation, the number of infertile survivors increases up to 30–40% [9,10]. Radiotherapy tends to be avoided in pediatric protocols (e.g., Hodgkin lymphoma) due to its long-term side effects and potential formation of secondary cancers. Among pediatric cancers treated with protocols using both chemo- and radio-therapy are the following: Wilm’s tumor, sarcomas, neuroblastoma, medulloblastoma, gliomas [11].

### 2.1. Chemotherapy

A list of some of the commonly used pediatric and AYA chemotherapy protocols is demonstrated in Table 1.

The evidence describing adverse effects on fertility is based on retrospective cohort studies. As treatment protocols often consist of polychemotherapy, the contribution of individual chemotherapeutic agents is often difficult. However, some protocols have been demonstrated to cause the highest risk [15]. Among chemotherapeutic agents, the alkylating agents (particularly procarbazine and cyclophosphamide) seem to cause the highest risk in a dose-related manner, with the highest doses causing the greatest probability of permanent amenorrhea [16,17]. Amenorrhea induced by chemotherapy treatment may be transient, and menstruation may return after several months post treatment [18]. The gonadotoxic effect of chemotherapy is affected by patients’ age, type of chemotherapy, dosage, number of cycles used. Even though the chemotherapeutic regimens could be modified to minimize their gonadotoxicity potential, the primary focus is to maximize the probability to cure patients’ primary disease [19].

Giving an accurate fertility prognosis before the start of treatment is often very difficult. Even if the initially chosen treatment protocol is associated with a low risk of gonadotoxicity (e.g., ABVD in Hodgkin lymphoma), in case of relapse a chemotherapy regimen of higher gonadotoxicity potential is used as a second line of treatment, significantly decreasing fertility potential [20]. Additionally, even in treatments with low-risk infertility potential, fertility may be impaired when pregnancy is delayed [21,22]. This should be especially taken into consideration in children and adolescents who have not yet started family planning.

### 2.2. Radiotherapy

Gonadal injury may also be caused by field radiation of total body, abdominal or pelvis by disruption of the function of the hypothalamic-pituitary axis, ovarian failure or direct damage to the uterus [23,24,25,26]. Oocytes are extremely sensitive to radiation, with an estimated dose of <2 Gy required to destroy 50% of primordial follicles [27]. The magnitude of the effect is related to dose, fractionation schedule, and age at the time of treatment. Uterine irradiation at any age causes an increased risk of miscarriage, premature delivery, low birth weight and maternal hemorrhage. Radiation induced damages to structures and vessels present in the pelvis and abdomen can potentially interfere with fetal growth either by constraining the developing pregnancy and/or by restricting vascular support to the growing fetus, resulting in lower birth weight [28,29].

Cranial irradiation may impair the functioning of hypothalamic-pituitary axis within the radiotherapy field and result in endocrine deficits. Radiotherapy of hypothalamic and pituitary regions frequently resulted in changes of prolactin concentrations and caused gonadotropin deficiency [30,31]. Moreover, irradiation of nasopharyngeal tumors was likely to cause secondary hypothalamic damage [31]. A study by Green et al. [32] demonstrated that the relative risk of miscarriage was increased in women who received cranial or craniospinal irradiation as well as in patients, in whom the ovaries were located within or near the radiation field or within 5 cm of the field edge. In Table 2 we demonstrate the use of different radiotherapy regimens in cancer treatment.

Older patients are more vulnerable to gonadotoxicity induced by radiation when compared to younger girls. It is due to the age-related decline of the oocyte population [17,19]. Gonadal injury can show clinical manifestations as ovarian hormone insufficiency (delayed or absent puberty, premature ovarian insufficiency, premature menopause) and/or infertility [5]. Premature ovarian insufficiency (POI) is characterized by the absence of menstrual cycles for ≥4 months and two elevated serum follicle-stimulating hormone levels in the menopausal range [34]. It is linked with either direct or indirect adverse effects of both chemo- and radiotherapy on the nonrenewable pool of primordial follicles within the ovary [35]. POI not only affects fertility, but has an impact on patient’s well-being. It is also associated with osteoporosis, cardiovascular disease and compromised sexual health [36,37]

### 2.3. Hematopoietic Stem Cell Transplantation (HSCT)

There is no doubt that HSCT impairs patients’ fertility. Studies have shown that POI occurs in as much as 90% of women undergoing bone marrow transplant for hematological malignancies. It is due to the fact that the regimens used for patients’ conditioning consist of treatment protocols with high gonadotoxicity potential that precede both auto- and allo-HCST [38,39]. Patients undergoing allo-HSCT have a higher risk for gonadal damage than patients undergoing auto-HSCT. Moreover, the risk of infertility is greater for postpubertal patients [40]. The risk for gonadal damage also depends on the drugs used as a part of chemotherapy protocol (higher risk for protocols using alkylating agents) and radiotherapy (total body irradiation, pelvic, fractionated doses) [40]. Allo-HSCT may can be complicated by acute or chronic graft versus host disease (GVHD) and require i.a. a long-term treatment with steroids, which could further increase the damage to the ovaries and increase the risk of infertility [41].

## 3. Fertility Preservation Methods

### 3.1. Fertility Preservation

Fertility preservation requires active measures to be taken before cancer treatment begins. Despite the pressure to begin cancer treatment as soon as possible, it is important to discuss fertility preservation options. As for adult women, there have been already invented established methods such as oocyte/embryo cryopreservation and ovarian tissue cryopreservation, which are being used extensively all over the world. In prepubertal girls the hypothalamus-pituitary-ovarian axis is inactive and there are no mature oocytes in the gonadal tissue, which causes a major challenge. All fertility preservation methods for prepubertal patients are considered to be either debatable or experimental [42,43]. In Table 3. we demonstrate the possible fertility preservation methods depending on the patients’ age and the possible time delay that could be offered for fertility preservation before the start of cancer treatment.

*Embryo cryopreservation* is the most established technique used in female fertility preservation. It requires at least one 10–14 days cycle of ovarian stimulation and a surgical procedure to harvest the oocytes. The developments of current ovarian stimulation methods allow to start the stimulation at any time of the menstrual cycle without affecting their efficacy (random start stimulation) [49]. Because of the inactive hypothalamic-pituitary-ovarian (HPO) axis in prepubertal girls, this method is not suitable for them. Furthermore, as ovarian stimulation may cause high serum estrogen levels, it should not be used in estrogen-sensitive cancers such as breast or endometrial cancer as it may promote their growth [50,51]. However, alternative and potentially safer protocols have been introduced for these patients including natural-cycle IVF (in vitro fertilization), without ovarian stimulation, stimulation protocols with tamoxifen alone or combined with gonadotropins as well as stimulation protocols with aromatase inhibitors to reduce the production of estrogen [52,53] The oocytes are fertilized with sperm either from partner or a donor and are frozen for later implantation. Surgery related complications (bleeding, scarring, anesthesia related) and ovarian hyperstimulation syndrome may occur leading to an elongated time before the beginning of cancer treatment.

*Oocyte cryopreservation*—as with embryo cryopreservation, oocyte cryopreservation requires ovarian stimulation and a necessary surgical procedure, however, it has approximately a 3–4 times lower success rate [12]. The mature oocytes are cryopreserved using slow freezing or vitrification methods; the second is preferred as it results in a better post-thaw survival rate [54,55]. This method is an attractive alternative for single women as it does not require the use of sperm. Both embryo and oocyte freezing are expensive (approximately $8000 U.S. Dollars per cycle) and storage fees need to be paid. The complications are similar as in embryo cryopreservation.

*Ovarian tissue cryopreservation* (OTC) is becoming an established method in adult patients; however, its use remains experimental in children and adolescents. The biggest study so far regarding the use of ovarian tissue cryopreservation in prepubertal children was conducted by Pivot et al. [56] who reported its use in 418 girls and adolescents. The first birth obtained after transplantation of cryopreserved ovarian tissue of a pre-pubertal patient was reported in 2015 by Demeestre et al. [57] With concern to the accepted age-related decline in the number of nongrowing follicles from birth, young girls undergoing high gonadotoxicity treatments, are potentially perfect candidates for this method. The procedure requires a surgical procedure (usually laparoscopy) to remove ovarian tissue and a whole ovary may be removed. It is possible to remove only a part of the ovary and frequently it is enough to remove only the cortex of approximately one third of the ovary as it includes a sufficient number of antral follicles [58]. A possible practice is to take multiple ovarian cortical strips from one ovary [59].

Different methods of ovarian tissue collection are available; either laparoscopic cortical strip or oophorectomy. This method allows minimal time delay in the start of cancer treatment as no ovarian stimulation is required. Moreover, it has no lower age limit as it can be used in prepubertal girls and adolescents. The youngest patient reported to undergo OTC was 3.5 months old [56]. Ovarian tissue cryopreservation enables preserving a large number of oocytes within primordial follicles and allows for spontaneous and repeated conception [20]. There are two methods for OTC: slow freezing and vitrification. A meta-analysis by Shi et al. [60] suggested that vitrification may be a more effective method for ovarian tissue cryopreservation than slow freezing and allow less primordial follicular DNA strand breaks and better preservation of stromal cells.

*Gonadal shielding during radiation therapy* is a method in which lead blocks are used to reduce the dose of radiation delivered to patients’ reproductive organs. Whenever possible, it should be used to provide ovarian protection, especially in young girls. Gonadal shielding, if possible, is indicated in patients receiving radiotherapy for cervical, vaginal, rectal, anal cancers, Hodgkin’s or non-Hodgkin’s lymphoma of the pelvical region or Ewing’s sarcoma of the pelvis [13]. In order to reduce the risk of ovarian irradiation, a free margin of minimum 2 cm should equal at least 2 cm to account for inner organ movement [13]. Shielding does not protect against gonadotoxic effects of chemotherapy and has a limited role when both chemotherapy and radiation are given [12,61]. In accordance to ESMO Clinical Practice Guidelines, gonadal shielding may be an alternative strategy for ovarian transposition, without creating the need of a surgical procedure in patients who require cancer treatment [13].

*Ovarian transposition* (oophoropexy) is the surgical repositioning of ovaries away from the radiation field in order to reduce the dosage of radiation that reaches the ovaries. Either one or two ovaries can be relocated behind the uterus, craniolaterally or under the diaphragm. The transposition should be performed just before the start of radiation to prevent the return of ovaries to the previous position; involves a surgical procedure; After the cancer treatment, the patient may require ovarian repositioning or IVF to conceive. It has approximately a 50% success rate as the procedure can fail due to altered ovarian blood flow and scattered radiation [62]. Ovarian transposition carries a risk connected to the surgical procedure, tube infarction, bleeding, pain. The efficiency of this method is controversial; as reported by Wo et al. the incidence of POI after ovarian transposition was found to be 50–90% [19]. It can be performed via mini-laparotomy, laparoscopy or robotic surgery [43]. Oophoropexy does not protect against gonadotoxic effects of chemotherapy [6,63].

*Ovarian suppression with gonadotropin releasing hormone (GnRH) analogs*—Numerous trials and meta-analyses have demonstrated a correlation between the use of GnRH analogs before and during chemotherapy treatment and lower incidence of premature ovarian insufficiency in young girls facing cancer [64,65,66,67,68,69] GnRH indirectly suppresses ovarian function by suppressing gonadotropin secretion from the pituitary gland [70,71]. The use of GnRH analogs in most cases does not protect against gonadotoxic effects of radiotherapy, especially if treatments associated with a high risk of gonadotoxicity are used such as preparation for hematopoietic stem cell transplantation [43,69]. Their usage in fertility preservation is still debatable. In accordance with ASCO and ESMO, other fertility preservation options should be used if available [6,7,63]. However recent publications suggest its utilization in patients with breast cancer [72].

### 3.2. Fertility Restoration

Upon the completion of oncological treatment patients may want to proceed with fertility restoration. Patients should be counselled for the feasibility and safety of fertility restoration procedures and pregnancy based on patient and disease/treatment related factors including patients’ age, medical history, type of cancer, type, dose and duration of cancer treatment, time interval since treatment completion, type of fertility preservation option used, contraindications to pregnancy and hereditary conditions [13].

The efficacy of fertility preservation using oocyte and embryo cryopreservation is tightly related to the number of mature oocytes received as the result of ovarian stimulation. In patients with low ovarian reserve presenting with low AMH levels, and high risk of POI, in cases when there is no urgent need to initiate anticancer treatment, double stimulation may be considered in order to increase the number of retrieved oocytes [73]. Successful cryopreservation of oocytes and embryos is essential to maximize the efficacy and safety of IVF treatment and to allow fertility preservation. Two methods are routinely used for oocyte, embryo and blastocyst cryopreservation: slow-freezing and vitrification. The systematic review and meta-analysis by Rienzi et al. suggest the superiority of vitrification to slow-freezing with regard to clinical outcomes and cryosurvival rates for oocytes, cleavage-stage embryos and blastocysts [74].

Ovarian in vitro maturation (IVM) could be a future method allowing for retrieval of immature oocytes from unstimulated ovaries, when ovarian stimulation is not possible, including prepubertal patients, patients with limited time for ovarian stimulation, when chemotherapy needs to be started immediately or those with a contraindication to sustained elevations of estradiol [75]. The harvested oocytes could be further cultured in vitro for 24–48 h to mature into metaphase II oocytes and to become ready to be used for IVF or vitrification [76,77,78]. In vitro oocyte maturation is an experimental method and is not commonly used, however, its success rates are improving and seem to have similar results as traditional IVF procedure [75,79,80,81,82]. IVM may be conducted along with ovarian tissue maturation [83]. Extracorporeal oocyte maturation is a possible way for future development of fertility preservation for prepubertal patients.

Currently, the only clinically available method that allows restoration of ovarian function and fertility is either orthotopic or heterotopic transplantation of cryopreserved ovarian tissue. In most patients the restoration of ovarian function was achieved within 4–9 months after ovarian tissue transplantation, however the restoration of ovarian function after grafting is variable in time duration and several graft procedures may be required to achieve pregnancy [83]. It should be noted that patients with a high risk of malignant contamination to the ovaries (e.g., aggressive forms of hematological malignancies) should not be eligible for ovarian tissue auto-transplantation [20,43] The perspectives for future research should include the emerging methods of in vitro maturation and in vitro culture of primordial follicles [84].

## 4. Oncofertility Management

Treatment of the primary disease is of the highest importance. Together with the cancer diagnosis, patient’s health status and staging of the disease should be assessed in order to complete patient’s evaluation and decide on the treatment protocol. The possible risk of gonadotoxicity should account for the proposed treatment (chemotherapy and/or radiotherapy), localization of the malignancy, dosage, number of cycles used, as well as additional factors including patients’ age and pre-treatment ovarian reserve that determine individual risk for immediate infertility or premature ovarian failure after the resumption of menses [12]. Fertility preservation should be discussed with the patients as early as possible to help with fertility preservation planning.

After the estimation of gonadotoxicity potential, a multidisciplinary team of specialists including oncologists, pediatricians, gynecologists, endocrinologists and psychologists should decide on the possible options of fertility preservation. Patients’ preferences and their willingness to have potential genetic children should be considered. In Figure 1 we demonstrate the steps required for a complex oncofertility management.

## 5. Psychological Aspects

Cancer diagnosis on its own is associated with emotional distress. A study by Lang et al. [85] demonstrated that AYA cancer survivors experience a significantly higher risk of psychosocial distress than their cancer-free peers and older adult cancer survivors. Moreover, cancer survivorship in AYAs was strongly associated with a higher prevalence of both mood and anxiety disorders than their cancer-free peers. Adding on top complications related to cancer treatment such as infertility often results in severe anxiety. Many adolescent and young adult cancer patients and survivors express a desire to have children in the future and worry about their fertility, regardless of their diagnosis, prognosis and form of treatment [86]. Fertility preservation is of great importance to many people diagnosed with cancer, especially to young patients who do not have children. Patients, who become infertile because of the treatment, have been identified to have an increased risk of emotional distress and lower long-term quality of life due to reduced life satisfaction, relationship problems, depression and increased anxiety [12,87,88]. The hope of being able to have a child after cancer treatment can contribute to a better acceptance of oncological therapy and its adverse effects, and improves the patients’ subjective experience of cancer treatments [89,90]. It is important to ask patients about a prospective desire to have children and to provide them with comprehensive information and explanation about the available fertility conservation methods.

Sullivan-Pyke et al. [91] conducted a study on the influences on the decision-making of prepubertal girls and young women undergoing ovarian tissue cryopreservation. The accepters as well as their parents (90.9% and 100% respectively) were driven by the desire for genetically related children and prevention of the stress of infertility. The decision to pursue fertility preservation is known to be difficult. It is even more complicated in the pediatric and adolescent population as unique ethical issues arise. The decisional conflict and regret may be greater for parents who have to decide for their child [92].

The distress and concern are related both to the later reproductive function and patient’s perception of fertility status and its impacts on psychological well-being during survivorship [89]. Patients should be offered psychological help during every stage of cancer treatment. Once being declared in remission, patients are often left alone, and only monitored once every couple of months. They are rarely asked how do they recover not only physically but get back to “normal” psychosocial functioning. Psychological help should be offered in this crucial time, to ease the stress and anxiety related to the side effects of cancer treatment.

Current advanced oncological treatment contributes to a high life expectancy and a great chance of cure of the primary disease, making it possible for cancer survivors to have their families and live many years post-treatment. Fertility preservation should be offered whenever possible, thus giving patients the opportunity to experience pregnancy and encourage positive thinking regarding survival.

Patients should also be offered psychological counseling when they start family planning many years post-treatment. It is especially important if they unsuccessfully try to conceive during a longer period of time or face miscarriages. They may also feel an increased fear related to invasive medical procedures and frequent medical appointments related to the assisted fertility procedures, e.g., IVF, which may resemble them of the past and frequent hospital visits related to the previous cancer treatment.

## 6. Ethical Considerations

Young girls, especially prepubertal, cannot fully understand the significance of fertility and its possible loss. Parents are the ones to decide for them, therefore assuming the possible future wishes to have a family. Comprehensive counseling considering prospects and risks of fertility preservation methods are required for patients and their supervisors. The risks of infertility vary depending on the type of treatment as well as patients’ age. Younger women are less likely to experience permanent amenorrhea than older patients, however, even if they continue to menstruate, they have a greatly increased risk of premature menopause and POI [93]. The informed consent process for minor patients requires the involvement of patient’s parents or legal guardians. An assent (a type of permission, less than full consent) is required in case of minors who are able to understand the issue. For children, who are too young to give assent, parents may consent to the experimental procedures only if the expected benefits are sufficient to justify the risks involved [94].

Fertility preservation techniques involve invasive procedures. They also carry an uncertain risk for tissue contamination in hematological and other malignancies. Decisions about undergoing any of the fertility preservation procedures should be made after an assessment of individual’s risk of fertility loss, based on patient’s staging and the protocol chosen. At the time of fertility preservation, the delay to cancer treatment, family’s comprehension that the process is experimental, surgical and anesthetic risks, child’s comfort. The ethical aspects that may arise in the future are: the impact of the surgery on gonadal function, tissue storage costs, reseeding of the original disease when tissue is reimplanted, the fate of tissue in case of death, false hope about the likelihood of pregnancy, the health of future offspring [95]. The concern about creation of false hope is particularly important in relation to prepubertal patients, where the likelihood of pregnancy in adulthood is remote and depends on future studies.

In adults and adolescents, the fertility preservation strategies are well established and many live births were reported. However, there is limited evidence for efficacy for tissue collected from prepubertal patients which raises an important ethical question if it is ethically justifiable to offer them fertility preservation methods involving surgical procedures [95]. No significantly increased risk for congenital anomalies or major mutagenic effects have been recorded in offspring born to patients successfully treated for cancer [96,97]; however, a concern may arise in patients with cancer-predisposing germline mutations. Some of the patients want to reproduce and have their own children only if they have an assurance that their children would not have a high risk to have cancer [98]. Identification of risk factors affecting fertility is important to provide patients with a high risk of gonadal dysfunction with proper counseling and referral to oncofertility teams for proper interventions. Pediatric oncofertility needs to be developed and in-depth information should be provided for patients, parents and their medical teams to facilitate the decision.

Upon the consideration of the implementation of fertility preservation, the benefits associated with the procedures should be considered. They provide a chance to have genetic children and become a parent in the future. The chance, especially for prepubertal patients, can be remote but are greater than if fertility preservation procedures were not undertaken. Even if the procedure is not successful, it demonstrates the concern for patient’s future fertility. Knowing that it was attempted, the patient may be comforted in adulthood to know that his parents considered this aspect of well-being.

The decision about the method of oncofertility preservation is often difficult make as procedures requiring ovarian stimulation and or surgical procedures may cause a delay in the start of cancer treatment. It is crucial that patients’ characteristics are evaluated before deciding on the method of fertility preservation. Additionally, if the patient agrees to go forward with the method involving ovarian stimulation, in order to reduce the time required for ovarian stimulation, the patient may undergo random-start controlled ovarian stimulation [99,100,101]. In cases when the start of cancer treatment is urgent, hormonal stimulation may be initiated at any time of the menstrual cycle. However, the amount and quality of follicles may be worse and result in lower rate of oocytes suitable for preservation [50].

## 7. Follow-Up and Survivorship Care

In Table 4 we demonstrate the possible complications that may occur as a result of chemotherapy and radiotherapy treatment.

Close monitoring of pubertal onset is recommended in girls who have received abdominopelvic radiotherapy and/or cytotoxic treatment. According to the COG Long-Term Follow-Up Guidelines for Survivors of Childhood, Adolescent, and Young Adult Cancer (COG-LTFU Guidelines) [103] yearly physical examination is recommended until reaching sexual maturity. Tanner staging, patients’ weight and height should be measured in order to exclude both precocious and delayed puberty. Medical history should be assessed yearly, asking for patients’ pubertal, menstrual and pregnancy history as well as sexual function. Patients should report any clinical symptoms such as hot flushes, lack of menstrual bleeding, irregular periods [104]. Laboratory screening of LH, FSH and estradiol levels should be conducted if clinically indicated. In case of abnormal hormone levels and clinical signs of hypogonadism or precocious puberty patients should be referred to endocrinology/gynecology specialists. In patients with delayed/arrested puberty, pubertal induction with sex steroid replacement therapy should be considered and closely monitored by pediatric endocrinologists and gynecologists [105].

Girls who underwent chemotherapy or abdominopelvic radiotherapy should monitor their ovarian function in adulthood. The counseling, regarding the risk of premature ovarian failure (POF), should include antral follicle count using transvaginal ultrasound as it is the most established method for assessing ovarian reserve. During the examination endocervix, endometrium and antral follicle count should be performed. Moreover, assessment of blood hormone levels (FSH, LH, estradiol) may be helpful in determining ovarian reserve; however, it remains difficult in prepubertal children as their hormone levels remain low [25,106] The hormone levels may be altered due to the chemotherapy/radiotherapy regimens used and normalize with time. It is essential to allow the hypothalamus ovarian axis to recover and await if the hormonal status comes back on its own with time before inducing sex steroid replacement therapy.

Anti-mullerian hormone (AMH) shows promise to be used as a hormonal marker of reduced ovarian reserve in women who have been treated for cancer [25]. It is also detectable in children [106]. AMH is produced in the ovaries by granulosa cells of primary, preantral and small antral follicles [107,108]. It indirectly reflects the quality and quantity of ovarian follicles at a given time predicting potential ovarian function. AMH is sensitive to changes accompanying with age; it peaks at approximately 25 years of age and becomes undetectable before menopause [109,110,111,112,113].

Even, if after the cancer treatment patients begin to menstruate regularly, they should be continuously monitored. They should be informed about the long-term gonadotoxicty potential of the cancer therapy and the possible risk of POI. Patients should be advised not to delay the motherhood as they may have lower ovarian reserve and an increased risk of POI. In the same time, they should wait with the pregnancy to allow the body to recover from the cancer treatment and deplete of any remaining chemotherapeutics that may alter fetal development. The first two years after the cancer treatment carry the highest risk of relapse, therefore patients should be advised to await this period, while constantly monitoring themselves and preferably plan the pregnancy after this time.

Preconception counseling should be offered and encouraged. Patients, who have obtained radiotherapy treatment to the pelvic region, including the uterus, have an increased risk of adverse outcome in pregnancy which includes late miscarriage, premature delivery, low birth weight. Patients should be advised that the pregnancy needs to be supervised in a referral, high-risk obstetric unit [105]. It is also important to include psychological counseling during the follow-up period. The counseling is especially important twice: first during fertility preservation and cancer treatment, and later during family planning which may result in unsuccessful attempts to get pregnant and miscarriages.

During the follow-up, gynecologists should notice the type of cancer treatment and method of fertility preservation that was offered to the patients. Patients who underwent ovarian stimulation procedures (sometimes multiple stimulation regimens) should be monitored as they may have an increased risk to develop secondary malignancy (borderline or invasive ovarian tumors) when compared to women unexposed to ovarian stimulating drugs [114]. Moreover, patients who received radiotherapy to pelvic and/or abdominal region should be monitored multidimensionally, not only towards premature ovarian failure but also minding the possibility of iatrogenic neo formation.

## 8. Conclusions

Cancer treatments have become more and more effective and the number of long-term survivals is rising. Infertility is a common long-term side effect of cancer treatment which can reduce patients’ quality of life. Growing population of pediatric and AYA cancer survivors, who are/or will be interested in having children, creates a need for fertility preservation methods that will allow similar success rates as in non-cancer population. Further research should be continued to improve the success rates of currently used fertility preservation methods as well as to develop new methods that will allow fertility preservation of even the youngest pediatric cancer patients. Moreover, as often an immediate start of antineoplastic treatment is required, methods that will allow the reduction of time needed for hormonal stimulation for oocyte maturation are of a great necessity. Fertility preservation options should be thoroughly discussed with patients and their families considering individual patients’ characteristics such as age, partner status, medical condition, urge for the start of cancer treatment and gonadotoxicity risk. Patients should be closely monitored after the end of cancer treatment to evaluate the risk of POI and long-term gonadotoxicity.

## Figures and Tables

**Figure 1 cancers-13-00202-f001:**
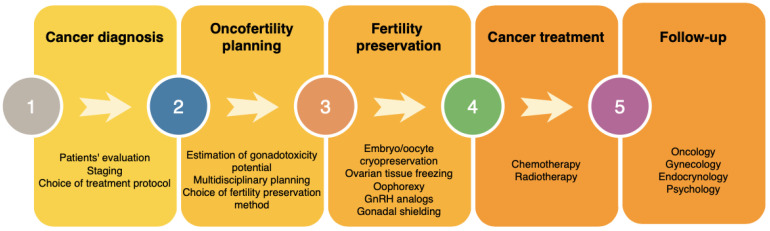
Oncofertility management of females diagnosed with cancer.

**Table 1 cancers-13-00202-t001:** List of chemotherapy protocols with gonadal toxicity potential based on ASCO (American Society of Clinical Oncology) recommendations on fertility preservation [12,13,14]. Note: Table contains examples and is not a complete list.

Degree of Risk of Permanent Amenorrhea	Chemotherapy Agent	Type of Cancer Treated
High (>80%)	Bone marrow transplant conditioning Cyclophosphamide > 7.5 g/m^2^ in females age < 20	Leukemias, lymphomas non-Hodgkin’s lymphoma, neuroblastoma, acute lymphoblastic leukemia, sarcoma
Intermediate (20–80%)	CAF (cyclophosphamide, doxorubicin, fluorouracil)	Breast cancer
	CEF (cyclophosphamide, epirubicin, fluorouracil)	Breast cancer
	CMF (cyclophosphamide, methotrexate, fluorouracil)	Breast cancer
	BEACOPP (doxorubicin, bleomycin, vincristine, etoposide, cyclophosphamide, procarbazine)	Hodgkin’s lymphoma
Low (<20%)	ABVD (doxorubicin, bleomycin, vinblastine, prednisone)	Hodgkin’s lymphoma
	AC (doxorubicin/cyclophosphamide)	Breast cancer
	CHOP (cyclophosphamide, doxorubicin, vincristine, prednisone)	Non-Hodgkin’s lymphoma
	Anthracycline/cytarabine	Acute myeloid leukemia
	CVP (cyclophosphamide, vincristine, prednisone)	Non-Hodgkin’s lymphoma, Follicular lymphoma
	MAP (cisplatin, doxorubicin, methotrexate)	Osteosarcoma
	VDC/IE (doxorubicin, vincristine, cyclophosphamide, ifosfamide, etoposide)	Ewing’s sarcoma
Very low or no risk	Methotrexate	Leukemia, head, neck, breast, lung cancer, rhabdomyosarcoma
	Fluorouracil	
	Vincristine	
	Tamoxifen	
Unknown risk (examples)	Monoclonal antibodies (trastuzumab, bevacizumab, cetuximab) Tyrosine kinase inhibitors (erlotinib, imatinib) Taxanes	Breast, ovarian, colon, lung cancer, glioblastoma, non-small lung cancer, pancreatic cancer, chronic myelogenous leukemia, acute lymphocytic leukemia, gastrointestinal stromal tumors, ovarian, breast, cervical, lung, pancreatic cancer

**Table 2 cancers-13-00202-t002:** Treatments considered to carry a high risk of radiotherapy related infertility or infertility-related outcome in women (>80%) [33].

Treatment	Type of Cancer Treated
Total body irradiation	Conditioning for bone marrow transplant (leukemias, lymphomas)
Pelvic and abdominal radiation ≥6 Gy in adult women ≥10 Gy in postpubertal girls ≥15 Gy in prepubertal girls	Ovarian cancer, endometrial cancer, cervical cancer, Wilm’s tumor, osteosarcoma, Ewing sarcoma, neuroblastoma, rhabdomyosarcoma

**Table 3 cancers-13-00202-t003:** Options for fertility preservation in girls undergoing cancer treatment [12,43,44].

Method		Can It Be Used in Prepubertal Girls?	Does It Cause Any Treatment Delay?	Does It Involve a Surgical Procedure?	Success Rate
Established methods	Embryo cryopreservation	No	Yes	Yes	Live birth rate 27.7% per frozen embryo [43]
	Oocyte cryopreservation	No	Yes	Yes	Live birth rate of 3–6% per frozen oocyte [43]
Experimental methods	Ovarian tissue freezing and transplantation	Yes	No	Yes	Live birth rate 32% per transplant; endocrine recovery rate was 93% [45]
	Oocyte in vitro maturation	Maybe	No	Yes	21.5–55.6% per cycle [46]
Debatable methods	GnRH analogs	No	No	No	Debatable
	Oophoropexy	Yes	No	Yes	66–79% of ovarian function preservation [47,48]
	Gonadal shielding	Yes	No	No	Debatable

**Table 4 cancers-13-00202-t004:** Female reproductive complications associated with cancer treatment [102,103].

Cancer Treatment	Complication	Risk Factors
Chemotherapy	Hypogonadism (gonadotropin deficiency, delayed or arrested puberty, acute ovarian failure, premature menopause, infertility)	High doses of alkylating agents, heavy metals and nonclassical alkylators; Combination of chemotherapy and radiation
Radiotherapy	Hypogonadism (gonadotropin deficiency, delayed or arrested puberty, acute ovarian failure, premature menopause, infertility)	Prepubertal gonadal irradiation ≥ 10 Gy Pubertal gonadal irradiation ≥ 5 Gy
	Precocious puberty	Young age at treatment, radiation dose ≥ 18 Gy to cranial regions
	Uterine vascular insufficiency	High pelvic radiation dose Radiation dose ≥ 30 Gy Patients with Wilms tumor and Mullerian anomalies
	Sexual dysfunction (vaginal fibrosis or stenosis)	Hypogonadism Graft-versus-host disease Prepubertal irradiation ≥ 25 Gy Postpubertal irradiation ≥ 50 Gy
Surgery	Sexual dysfunction (vaginal fibrosis or stenosis)	Spinal cord tumors, vaginal tumors, surgery involving pelvical region

## Data Availability

Not applicable. No new data were created or analyzed in this study.

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
