# Peer review of "Fertility Preservation and Long-Term Monitoring of Gonadotoxicity in Girls, Adolescents and Young Adults Undergoing Cancer Treatment"

_cancers, 2021, doi:10.3390/cancers13020202_

Round 1
Reviewer 1 Report
In their review “Fertility preservation and long-term monitoring of gonadotoxicity in girls, adolescents and young adults undergoing cancer treatment.”, the authors make an updated inventory of girls and young females facing cancer therapy.
The review is very well written and offers an up-to-date view of articles on the field. In addition, many parts (sometimes put aside or very little detailed in this thematic) are covered and well presented: Psychological Aspects and Ethical Considerations for pediatric and adolescent patients.
However, there are some minor changes to be made to the review.
Comments
- The part “Fertility preservation methods” could be more developed. Indeed, it seems essential to have two parts (or sub-parts?) considering the phases of fertility preservation (cryopreservation) and fertility restoration (tissue grafting, cell transplantation, in vitro maturation, etc…). Moreover, without needing to go into detail, it seems important that these sub-parts highlight the different existing possibilities for (i) the fertility preservation (controlled or uncontrolled slow freezing, vitrification) for cells but also for ovarian tissues and (ii) the restoration of fertility (where are we in humans? Is in vitro oogenesis a way forward?). The section “Ovarian tissue cryopreservation and transplantation” is probably to be detailed a little more in this sense.
- The sub-part “Gonadal shielding during radiation therapy” deserves to be explained a little more clearly.
- A ‘Conclusion Part’ could be beneficial. Indeed, one has the impression that the review is ending without starting on a possible step forward in what will be possible in the future.
Specific to the Figure
- Page 7, Figure 1. Please, increase the font for the numbers (1 to 5) as well as the font for the descriptions below the ‘square’ representations. In addition, please reduce the upper descriptions (and lower them a little so that, in particular for the ONCOFERTILITY). Moreover, capitalizing of everything remains a strange choice.
Specific to the Table
- Page 2, Table 1. Please, check that the CAF (cyclophosphamide, doxorubicin, fluorouracil) for Breast cancer was not has not been duplicated by mistake (or that another chemotherapy agent is not missing instead for Intermediate (20-80%) degree of risk of permanent amenorrhea).
- Page 2, Table 1. Please, replace “ammenorrhea” by “amenorrhea”.
Specific Minor Comments
- Page 1, Line 41. Please, specify the abbreviation ‘CNS’ (central nervous system)
- Page 2, Line 41. Please, replace [2], [3], [4] by [2–4]. Indeed, please follow the instructions of MDPI concerning the references number throughout the entire manuscript: “In the text, reference numbers should be placed in square brackets [ ], and placed before the punctuation; for example [1], [1–3] or [1,3].”.
- Page 2, Line 61. Please, check that there is no additional space before “The long-term effects”.
- Page 2, Line 73. Please, replace [9], [10] by [9–10].
- Page 3, Line 87. Please, replace [14], [15] by [14–15].
- Page 3, Line 95. Please, check that there is no additional space between “a higher toxicity potential is” and “used”.
- Page 3, Lines 94 to 96. Could the authors reword the paragraph to help understanding? Indeed, it is difficult to perceive the meaning of this sentence.
- Page 3, Line 97. Please, replace [19], [20] by [19–20].
- Page 3, Line 101. Please, replace [21]–[24] by [21–24].
- Page 3, Line 108. Please, replace [26], [27] by [26–27].
- Page 3, Line 112. Please, replace [28], [29] by [28–29].
- Page 4, Line 130. Please, replace [35], [36] by [35–36].
- Page 4, Line 135. Please, suppress the space between “oocyte/” and “embryo”.
- Page 4, Line 137. Please, check that the “ova” is not a typo.
- Page 4, Line 139. Please, replace [37], [38] by [37–38].
- Page 5, Lines 149 to 150. Please, replace [46], [47] by [46–47].
- Page 5, Line 156. Please, replace [50], [51] by [50–51].
- Page 5, Line 156. To be consistent with others paragraphs, please replace “- It” by “is becoming”.
- Page 6, Line 200. Please, replace the “;” by a “.” or rephrase the sentence.
- Page 6, Line 208. Please, replace [58], [59] by [58–59].
- Page 6, Line 212. Please, replace [60]–[64] by [60–64].
- Page 6, Line 214. Please, replace [65], [66] by [65–66].
- Page 6, Line 216. Please, replace [67], [68] by [67–68].
- Page 6, Line 218. Please, replace “–” by “-”.
- Page 6, Line 223. Please, replace [71]–[73] by [71–73].
- Page 6, Line 223. Please, check that there is no additional space before “In”.
- Page 6, Line 226. Please, replace [74]–[78] by [74–78].
- Page 6, Line 231. Please, suppress the space between “and/” and “or”.
- Page 6, Lines 240 to 241. Please, suppress the “.” and replace the “We” by a “we” or rephrase the sentence. In addition, please check that there is no additional space between “complex oncofertility” and “management”.
- Page 7, Line 246. Please, check that there is no additional space before “demonstrated”.
- Page 7, Line 257. Please, replace [82]–[84] by [82–84].
- Page 7, Line 260. Please, replace [85], [86] by [85–86].
- Page 7, Line 264. Please, add a space between “cryopreservation” and “(OTC)”. In addition, why the authors whose to use an abbreviation of this term only page 7?
- Page 9, Line 341. Please, replace [95]–[97] by [95–97].
- Page 10, Line 341. Please, replace [106], [107]–[110] by [106–110].
- Page 10, Lines 383 to 384. Please, replace “premature ovarian insufficiency (POI)” by “POI” (who was already presented before).
Author Response
Dear reviewer,
Thank you for your comments. We believe they add great value to the manuscript.
Comments
- The part “Fertility preservation methods” could be more developed. Indeed, it seems essential to have two parts (or sub-parts?) considering the phases of fertility preservation (cryopreservation) and fertility restoration (tissue grafting, cell transplantation, in vitro maturation, etc…). Moreover, without needing to go into detail, it seems important that these sub-parts highlight the different existing possibilities for (i) the fertility preservation (controlled or uncontrolled slow freezing, vitrification) for cells but also for ovarian tissues and (ii) the restoration of fertility (where are we in humans? Is in vitro oogenesis a way forward?). The section “Ovarian tissue cryopreservation and transplantation” is probably to be detailed a little more in this sense.
We divided the paragraph into two parts: 3.1 concerning fertility preservation and 3.2 concerning fertility restoration. We added the information on possible cryopreservation techniques. Moreover, we moved the paragraph on in vitro maturation to paragraph 3.2. We have also provided additional information on possible future research. Please see the improved version of the manuscript.
- The sub-part “Gonadal shielding during radiation therapy” deserves to be explained a little more clearly.
we improved the paragraph on gonadal shielding in accordance to ESMO guidelines; please see the improved version
- A ‘Conclusion Part’ could be beneficial. Indeed, one has the impression that the review is ending without starting on a possible step forward in what will be possible in the future.-
we added a general conclusion paragraph with an emphasis on the development of more sufficient methods with better success rates that will require less time and allow fertility preservation even for young pediatric patients
Specific to the Figure
- Page 7, Figure 1. Please, increase the font for the numbers (1 to 5) as well as the font for the descriptions below the ‘square’ representations. In addition, please reduce the upper descriptions (and lower them a little so that, in particular for the ONCOFERTILITY). Moreover, capitalizing of everything remains a strange choice.
we improved the figure, please see the revised version
Specific to the Table
- Page 2, Table 1. Please, check that the CAF (cyclophosphamide, doxorubicin, fluorouracil) for Breast cancer was not has not been duplicated by mistake (or that another chemotherapy agent is not missing instead for Intermediate (20-80%) degree of risk of permanent amenorrhea).
thank you for noticing, we changed it into CMF which was missing
- Page 2, Table 1. Please, replace “ammenorrhea” by “amenorrhea”.
thank you for noticing the mistake, we corrected the spelling
Specific Minor Comments
Thank you for your comments, we developed the abbreviations used, corrected spelling mistakes, formatting and bibliography
Reviewer 2 Report
The paper is very well written, and reports the latest evidences on a very sensitive aspect of comprehensive care for oncology patients.
I have only a few minor suggestions:
1. Table 1 reports the recommendations of ASCO and I noticed that most typical pediatric cancers are not included; I guess this is because pediatric neoplasias are a minority of all cancers and also because available evidence on fertility issue in patients of very young age and affected by a pediatric cancer are very scarce. Although thr latter consideration is very well explained in the thext, table 1 could be misinterpreted and lead to the conclusion that a patient affected by a neuroblatoma is not at risk of infertility. Therefore, I suggest to either review the list or make it clear that the list is far from being complete
2. When considering "agents affecting feritlity" a special mention should be reserved to Stem Cell Transplant
3. Lines 209-217, Ovarian suppression with gonadotropin releasing hormone (GnRH) analogs; the Authors should include another reference: “Meli M, Caruso-Nicoletti M, La Spina M, Nigro LL, Samperi P, D'Amico S, Bellia F, Miraglia V, Licciardello M, Cannata E, Marino S, Cimino C, Puglisi F, Valvo LL, Pezzulla A, Russo G, Di Cataldo A. Triptorelin for Fertility Preservation in Adolescents Treated with Chemotherapy for Cancer. J Pediatr Hematol Oncol 2018, 40(4):269-76. doi:10.1097/MPH.0000000000001144”.
Author Response
Dear reviewer,
Thank you for your comments. We believe they add great value to the manuscript.
- Table 1 reports the recommendations of ASCO and I noticed that most typical pediatric cancers are not included; I guess this is because pediatric neoplasias are a minority of all cancers and also because available evidence on fertility issue in patients of very young age and affected by a pediatric cancer are very scarce. Although thr latter consideration is very well explained in the thext, table 1 could be misinterpreted and lead to the conclusion that a patient affected by a neuroblatoma is not at risk of infertility. Therefore, I suggest to either review the list or make it clear that the list is far from being complete
We were debating on the outlook of the table for a long time. According to different guidelines, different authors show different gonadotoxicity potential for different therapies. In some manuscripts, some therapies are listed as medium risk whereas in others are listed as low risk. We know, there are many chemotherapy treatment protocols that carry a risk of gonadotoxicity however, as we couldn’t clearly agree on their gonadotoxicity potential, we decided to include only a few of them, listed in ASCO guidelines, which are used for some of the most common malignancies of pediatric and adolescent age. Additionally, we decided to add information on protocols used for rhabdomyosarcoma, neuroblastoma and Ewing sarcoma. We have also added the information that the table contains examples and is not a complete list
2. When considering "agents affecting feritlity" a special mention should be reserved to Stem Cell Transplant - we added a paragraph on stem cell transplant. Please see the improved version of the manuscript
3. Lines 209-217, Ovarian suppression with gonadotropin releasing hormone (GnRH) analogs; the Authors should include another reference: “Meli M, Caruso-Nicoletti M, La Spina M, Nigro LL, Samperi P, D'Amico S, Bellia F, Miraglia V, Licciardello M, Cannata E, Marino S, Cimino C, Puglisi F, Valvo LL, Pezzulla A, Russo G, Di Cataldo A. Triptorelin for Fertility Preservation in Adolescents Treated with Chemotherapy for Cancer. J Pediatr Hematol Oncol 2018, 40(4):269-76. doi:10.1097/MPH.0000000000001144”. - thank you for your suggestion; we added this reference
Reviewer 3 Report
This paper is an interesting and complete review about fertility preservation in girls with cancer treated with chemotherapy, radiotherapy and surgery.
In my opinion the subject is well analyzed in all its parts. Just a few suggestions:
31-32: no parenthesis
Table 1: to be completed with pediatric protocols for rhabdomyosarcoma, neuroblastoma and Ewing sarcoma
99: add cranial irradiation
Table 2: add Ewing sarcoma
151: explain IVF
188: explain OTC
212: most of references 60-64 concern studies with adult patients. Please add: Meli M, Caruso-Nicoletti M, La Spina M et al. Triptorelin for Fertility Preservation in Adolescents Treated With Chemotherapy for Cancer. J Pediatr Hematol Oncol 40:269-276,2018
214-215: reference 38 is not enough to be so absolute, please add something like "probably" "in most cases"
232: its is not correct. It is not the site of the treatment, but the site of the tumor
Figure 1: in the second rectangle correct multidisciplinary
Author Response
Dear reviewer,
Thank you for your comments. We believe they add great value to the manuscript.
Table 1: to be completed with pediatric protocols for rhabdomyosarcoma, neuroblastoma and Ewing sarcoma we added the mentioned protocols to the table
99: add cranial irradiation - we were thinking about including a part on cranial irradiation, however as cranial irradiation does not cause permanent infertility (hormonal replacement therapy can be used) we decided not to include this information in the table; however, we mention the effect of cranial irradiation on patients’ infertility in the main text.
Table 2: add Ewing sarcoma- thank you, we added Ewing sarcoma to the table
151: explain IVF
188: explain OTC- thank you, we have explained both of the abbreviations in the main text
212: most of references 60-64 concern studies with adult patients. Please add: Meli M, Caruso-Nicoletti M, La Spina M et al. Triptorelin for Fertility Preservation in Adolescents Treated With Chemotherapy for Cancer. J Pediatr Hematol Oncol 40:269-276,2018 - thank you for your suggestion; we added this article as a reference
214-215: reference 38 is not enough to be so absolute, please add something like "probably" "in most cases” - - thank you for your suggestion; we added “in most cases”
232: its is not correct. It is not the site of the treatment, but the site of the tumor- we changed it into “localization of the malignancy”
Figure 1: in the second rectangle correct multidisciplinary - thank you for noticing, we improved the figure as it required some additional visual changes